# Sepsis Mortality Prediction Using Wearable Monitoring in Low–Middle Income Countries

**DOI:** 10.3390/s22103866

**Published:** 2022-05-19

**Authors:** Shadi Ghiasi, Tingting Zhu, Ping Lu, Jannis Hagenah, Phan Nguyen Quoc Khanh, Nguyen Van Hao, Louise Thwaites, David A. Clifton

**Affiliations:** 1Department of Engineering Science, University of Oxford, Oxford OX3 7DQ, UK; tingting.zhu@eng.ox.ac.uk (T.Z.); ping.lu@eng.ox.ac.uk (P.L.); jannis.hagenah@eng.ox.ac.uk (J.H.); david.clifton@eng.ox.ac.uk (D.A.C.); 2Oxford University Clinical Research Unit, Ho Chi Minh City 710400, Vietnam; khanhpnq@oucru.org (P.N.Q.K.); lthwaites@oucru.org (L.T.); 3Hospital of Tropical Diseases, Ho Chi Minh City 700000, Vietnam; dr_nguyenvanhao@ump.edu.vn

**Keywords:** sepsis, wearable sensors, machine learning, low–middle income countries, resource-limited, continuous physiological signals, Vietnam, electrocardiogram, heart rate variability

## Abstract

Sepsis is associated with high mortality—particularly in low–middle income countries (LMICs). Critical care management of sepsis is challenging in LMICs due to the lack of care providers and the high cost of bedside monitors. Recent advances in wearable sensor technology and machine learning (ML) models in healthcare promise to deliver new ways of digital monitoring integrated with automated decision systems to reduce the mortality risk in sepsis. In this study, firstly, we aim to assess the feasibility of using wearable sensors instead of traditional bedside monitors in the sepsis care management of hospital admitted patients, and secondly, to introduce automated prediction models for the mortality prediction of sepsis patients. To this end, we continuously monitored 50 sepsis patients for nearly 24 h after their admission to the Hospital for Tropical Diseases in Vietnam. We then compared the performance and interpretability of state-of-the-art ML models for the task of mortality prediction of sepsis using the heart rate variability (HRV) signal from wearable sensors and vital signs from bedside monitors. Our results show that all ML models trained on wearable data outperformed ML models trained on data gathered from the bedside monitors for the task of mortality prediction with the highest performance (area under the precision recall curve = 0.83) achieved using time-varying features of HRV and recurrent neural networks. Our results demonstrate that the integration of automated ML prediction models with wearable technology is well suited for helping clinicians who manage sepsis patients in LMICs to reduce the mortality risk of sepsis.

## 1. Introduction

Sepsis is a life-threatening condition involving organ dysfunction in response to infection, and is a major global health concern [1]. The most recent global estimates for sepsis incidence and mortality reported 48.9 million sepsis cases and 11 million sepsis-related deaths, accounting for 19.7% of all global deaths in 2017 [2]. These statistics varies substantially across regions, with the highest burden reported in low–middle income countries (LMICs) in Sub-Saharan Africa, Oceania, East Asia, and Southeast Asia [3,4].

Despite the seriousness of sepsis as a public health problem in LMICs, the majority of studies target the sepsis population in clinical settings from high-income countries (HICs), further augmenting the skewed generation of medical knowledge, providing evidence and insights that cannot necessarily be generalised to under-represented populations [5].

Early recognition and treatment of sepsis are the two essential elements of improving outcome from the condition [2]. This requires careful clinical evaluation and monitoring of patients’ vital signs, with or without supporting laboratory data. Lack of laboratory access, monitoring equipment, and skilled health care staff are serious impediments to achieving this in LMICs [6,7,8]. To assist triage and prognostication, various scores have been developed. Those most suited to LMIC settings are those composed of simple-to-measure vital signs alone such as quick sequential organ failure assessment score (qSOFA) or National Early Warning Score (NEWS). However, their performance is sub-optimal and they still require staff to carry out observations regularly [9,10].

The application of novel digital health technologies may provide a solution to this [11,12]. Wearable sensors are innovative, cost-effective methods that provide continuous monitoring and an objective measure of the physiological status of critically ill patients [13]. Although such systems are often developed and validated in stable, ambulatory patients in HICs [14,15,16], their small, lightweight, and low-cost characteristics compared to traditional costly bedside monitors have increased their acceptability for use in LMICs [17]. Furthermore, the integration of automatic prediction algorithms means wearable systems could be especially attractive in settings with limited staff. Despite the great potential of wearable sensors in resource-limited healthcare systems, very few studies have conducted research on the use of wearable systems in these settings for critically ill patients [18,19].

The most available wearable sensors are pulse oximetry through pulse plethysmography (PPG) and those measuring the electrocardiogram (ECG). These may be of particular use for sepsis management as, in addition to oxygen saturation and heart rate, through detection of alterations in beat-to-beat PPG or ECG recordings, the heart rate variability (HRV) series can also be calculated. Changes in HRV have been demonstrated to be of prognostic value in sepsis and have been correlated with the outcome and the severity of the disease [16,20]. Hitherto, studies have used traditional “Holter” type monitors or data extracted from beside monitors to calculate HRV parameters, providing robust data for research but not practical for clinical practice [21].

Traditionally, conventional statistical techniques have been used to build prognostic prediction models in sepsis; however, biomarkers obtained from HRV can be embedded into automatic machine-learning (ML)-based prediction models, providing accurate predictions of a patient’s physiological state. ML models have been shown to outperform both internal decisions made by clinicians and clinical risk scores in predicting mortality in patients with sepsis [22,23].

However, the majority of these models i) use clinical, laboratory, vital sign observations, and physical data collected from traditional bedside monitoring systems, and ii) are trained and validated on the data gathered from HICs. To the best of our knowledge, no other study has integrated ML models with wearable monitoring systems in critically ill sepsis patients in LMICs. Therefore, in this study, we propose an innovative interpretable automated prediction algorithm integrated into wearable sensing devices for predicting the mortality rate of critically ill sepsis patients in LMICs. We aim to investigate the potential of HRV measures obtained from wearable sensors for the automatic prediction of in-hospital mortality at early admission stage. To achieve this, in our prospective study, we monitored the physiological status of adult sepsis patients admitted to the Hospital for Tropical Diseases (HTD) in Vietnam by acquiring the ECG signal using wearable sensors. The patients’ primary vital signs (temperature, pulse, systolic blood pressure, oxygen saturation) were also monitored through bedside monitoring systems. We designed ML models to automatically predict the hospital discharge outcome of patients after nearly two weeks from their hospital admission using data collected during their first day of admission. We compared the performance of the state-of-the-art ML models in predicting the hospital discharge outcome using the primary vital signs and HRV measures extracted from the wearable sensor data. We also provide interpretability analysis of the ML models to identify the most informative inputs, further obtaining a better understanding of the behaviour of each model which is helpful for the clinicians.

The three major contributions of this work are listed as follows:We gathered a rich dataset of simultaneous long-term ECG recordings from wearable sensors and vital signs from bedside monitoring systems from adult patients with sepsis admitted to the hospital within a resource-limited healthcare system.We demonstrate the high potential of physiological data gathered from wearable devices compared to traditional bedside monitoring systems for sepsis management.We propose an interpretable automatic ML-based solution for a long-term prediction of hospital discharge outcome in critically ill sepsis patients using the data collected during the first day of admission to be potentially implemented in practice.

The main novelty of our study is the design of an automatic mortality prediction ML-based pipeline using wearable data acquired from sepsis patients in LMICs. Our models are trained on a novel dataset comprising simultaneous recording of bedside monitor variables and wearable data collected from critically ill sepsis patients in LMIC settings in contrast to prior studies which have been mostly conducted in HICs. Moreover, our prediction pipeline offers an explanation of the predictions made by all the ML models, highlighting the importance of input variables for clinicians which is not usually studied in previous works.

## 2. Related Works

Monitoring physiological status of patients is crucial in modern critical care. Currently, the waveform displays from bedside monitors such as the ECG signal, respiration activity, and the time-averaged data extracted from these waveforms (such as heart rate) have been the basis for clinical decision-making [24]. However, recent developments in easy-to-use wearable devices have facilitated continuous long-term monitoring of physiological signals such as ECG or PPG signals [25]. These technologies augmented with interpretable and accurate artificial-intelligence-based models offer the prospect of not only reducing the cost of monitoring, but also improving patient management through data interpretation and clinical decision support, particularly in low-resource settings [26].

A study undertaken in Rwanda has shown the feasibility and high accuracy of wearable biosensor devices in monitoring vital signs of acutely ill paediatric and adult emergency department (ED) patients with sepsis in LMIC settings [19]. The authors demonstrated that vital sign measurements from a wireless wearable device are reliable and accurate compared to those obtained by an experienced nurse. Our previous work has also shown that the wearable devices are reliable and robust, and can be used as a surrogate for bedside monitors [27]. However, no previous attempt has been made to design automatic prediction algorithms based on wearable data in LMICs.

ML models have been trained on various types of data in sepsis population in different clinical settings for mortality prediction. These models have outperformed traditional risk stratification tools based on clinical scoring [22,23,28,29,30]. A recent study [22] used individual patient data containing clinical and laboratory information available within two hours after initial ED presentation as an input to ML-based models, obtaining an area under the curve of precision recall curve of 0.82, which outperformed the other clinical emergency department scoring systems.

In addition to clinical and laboratory data, HRV has been considered as a predictor of sepsis mortality in many studies, as reported by a systematic review [20]. Studies have reported reductions in several HRV parameters in septic patients who died [20]. However, all of these studies have implemented traditional, statistical methods to compare HRV parameters in survivors and non-survivors in the sepsis patients.

ML models trained on HRV extracted from 5 min ECG tracings performed at triage have been shown to improve the prediction of mortality in the suspected sepsis patients in ED compared to traditional risk stratification tools [23]. HRV measures have also been fused with other clinical and laboratory information recorded within 1 h of ED presentation, achieving higher performance for quantifying the risk of deterioration in sepsis patients [31].

Most of the studies [32,33,34] that used HRV data for sepsis mortality prediction considered short ECG recordings (10–30 min), and those with longer recordings (24–48 h) used Holter equipment for monitoring the ECG [21,35,36]. Whilst providing good quality research data, neither Holter recording nor data extraction from bedside monitors are feasible for routine clinical use, nor readily available in low-resource settings. This limits the generalizability of most of the previous studies in LMICs, where the healthcare sector is understaffed and digital infrastructure is underdeveloped. To the best of our knowledge, there is no study that used HRV data from wearable sensors for in-hospital mortality prediction in sepsis patients.

## 3. Materials and Methods

### 3.1. Study Participants

All adult patients with a clinical sepsis diagnosis who were admitted to HTD, Ho Chi Minh City in Vietnam, were screened for inclusion in this study. The hospital is a tertiary referral centre for infectious diseases serving Southern Vietnam. Sepsis was defined according to the HTD guidelines which are applicable to the local clinical situation at HTD. Septic shock was defined based on Sepsis-3 guidelines [1]. These include proven or suspected community-acquired bacterial infection, plus the sequential organ failure assessment score (SOFA) more than or equal to 2 plus persistent hypotension requiring vasopressors to maintain a mean arterial pressure (MAP) of >65 mm Hg and having a serum lactate of >2 mmol/L despite adequate volume resuscitation. The presumed source of infection was recorded for all patients, with supporting microbiology where available.

Exclusion criteria included history of allergy to electrodes, failure to give informed consent, and contraindications to the use of wearable sensors. As a result, 50 patients met the inclusion criteria and were recruited for this study. All patients gave informed consent and the study was evaluated to impose no risk to the patients. Our study was approved by the Scientific and Ethical Committee of the HTD, Ho Chi Minh City in Vietnam with the protocol number 1009/BVBND-HDDD and Oxford Tropical Research Ethics Committee with the protocol number 522-20.

### 3.2. Experimental Protocol (Data Collection)

The patients were recruited for this study within 24 h after their admission to the HTD. On enrolment, patient baseline clinical information was recorded. We used the ePatch^®^ ECG patch monitor (Delta Electronics, Denmark) as a wearable biosensor. The ePatch^®^ is a CE-marked, three-lead sternal ECG recording device capable of recording two channels of ECG data continuously for 24 h. It is lightweight and adheres to the patient’s chest. This device records single-channel ECG output with a sampling frequency of 256 Hz which is stored in the device and exported at the end of the recording period. The ePatch^®^’s position was determined according to its ideal position on the patient left chest—the highest edge of the sensor on the mid-line, about 4 cm from the left collarbone. The area was cleaned before attaching the patch by soap or alcohol pad. The sensor was tightly attached to the patch. The device was attached to the patient’s chest and continuous ECG data recording took place over a 24 h period. Meanwhile, the patients underwent routine clinical measurements every 1–6 h during their hospital stay, depending on clinical need, as part of the routine medical care. These include the collection of vital signs from bedside monitors using the GE CARESCAPE B450 patient monitor.

Once the data collection was completed with the ePatch^®^, it was slowly removed from the patient’s skin and the sensor was detached from the patch. The ePatch^®^ sensor was connected to the computer with its accompanying cable and the ECG data were stored for further analysis. At the hospital discharge, clinical outcome data together with hospital length of stay were reported. This is a prospective study which started on June 2020. The training procedure is performed offline after the training data are stored for all the patients.

### 3.3. Final Cohort of Patients

Out of 50 patients, the hospital discharge outcome for 9 patients was missing and therefore they were excluded for the analysis. Bedside monitoring measurements were missing for one patient. Therefore, the final dataset consisted of 40 patients. The demographic information of patients used for further analysis is reported in Table 1. This table includes patient’s characteristics, including age, sex, and hospital length of stay for survivals and non-survivals.

### 3.4. Processing Pipeline

A complete illustration of the processing pipeline is depicted in Figure 1. It consists of the following processing blocks.

#### 3.4.1. Data Collection and Feature Extraction

Bedside monitor: Vital signs comprising body temperature, pulse rate, systolic blood pressure, respiratory rate, and oxygen saturation were extracted from bedside monitors depending on their frequency of collection for each patient. These measurements were collected on an hourly basis for the majority of patients. For those patients who did not have regularly hourly vital signs, we interpolated their vital signs on hourly basis based on nearest past value. This choice was to synchronize the collection of vital signs with the monitored ECG signals on hourly basis for implementation purposes. Wearable monitor: We applied standard preprocessing methods to the single lead ECG signals acquired from the ePatch^®^. Each signal was normalized and filtered within ECG frequency bands ([0–4 Hz]) using Butterworth band-pass filter algorithm. Each ECG signal was segmented to hourly bins for further processing. The choice of hourly bins is to match wearable recordings with the vital signs. Within each window, the HRV signal was derived by extracting the R-peaks from the ECG signals using using the standard Pan–Tompkins algorithm [37]. Figure 2 shows an exemplar ECG signal acquired from the ePatch^®^, the detected R peaks, and the corresponding RR interval from 1 min acquisition of a random sepsis patient. We used the open-source packages *NeuroKit2* and *pyHRV* packages for the calculation of standard HRV parameters from time domain, frequency domain, and nonlinear domain [38,39].

The list of all features considered in this study along with their definition are described in Table 2. We also depicted dynamic trends of all the features along 24 h for an exemplar patient in Figure 3.

#### 3.4.2. Machine Learning Training

In order to train the ML models for the task of predicting in-hospital mortality using multivariate time series data from sepsis patients, we considered different modalities of information according to the following feature subsets.

$Sep_{HRV}$ = [HR(mean), HR(std),RR(mean), RR(std), RMSSD, $P_{VLF}$, $P_{LF}$, $P_{HF}$, $Peak_{VLF}$, $Peak_{LF}$, $Peak_{HF}$, SD1, SD2, SD1/SD2, SampEn, $\alpha_{1}$, $\alpha_{2}$]$Sep_{Vital}$ = [Temp, Pulse, SBP, Resp, SP02]$Sep_{HRVital}$ = [$Sep_{HRV}$,$Sep_{Vital}$]

Then, we constructed two different datasets from these feature subsets for a binary prediction task. For both datasets, the prediction output is hospital discharge outcome recorded at the discharge (on average two weeks after the patient’s commencement of monitoring); this is coded as positive in the case of a patient’s death, or negative in the case of a patient’s survival.

Taking into account the time-varying dynamics of the feature sets, we define a dataset as:

$D_{sep,t}=\left\{\left(X^{\left(p\right)},y^{\left(p\right)}\right)\;|\;X^{\left(p\right)}=\left[X_{lt}^{\left(p\right)}\right],y^{\left(p\right)}\in \left\{0,1\right\},\;p=1,\ldots ,P\;,\;l=1,\ldots ,L,t=1,\ldots ,T\right\}$ where *P* represents the number of sepsis patients, *L* is the total number of the time series features and *T* is the time duration of each time series. The *T* value is 24 since our observations are collected on hourly time bins along the first 24 h of hospital admission. Therefore, each $X^{\left(p\right)}=\left[X_{lt}^{\left(p\right)}\right]$ represents a rectangular matrix of size L×T for patient *p* and $y^{\left(p\right)}$ represents the class label, i.e., the hospital discharge outcome of each patient.

We also considered another dataset consisting of averaged dynamics of each multivariate feature along the time.

$D_{sep}=\left\{\left(x^{\left(p\right)},y^{\left(p\right)}\right)\;|\;x^{\left(p\right)}=\left[x_{1}^{\left(p\right)},\ldots ,x_{l}^{\left(i\right)}\right],y^{\left(p\right)}\in \left\{0,1\right\},\;p=1,\ldots ,P\;,\;l=1,\ldots ,L\right\}$, where
(1)$$\begin{array}{cc} x^{\left(p\right)} & =\frac{\sum_{t=1}^{T}\left[X_{lt}^{\left(p\right)}\right]}{T} \end{array}$$
and *p* and *L* are defined as above.

Based on each dataset’s architecture, we chose appropriate state-of-the-art ML models based on their performance and interpretability of the predictions [40]. Using the $D_{sep}$, we applied the following machine learning models.

Support vector machines with recursive feature elimination (SVM-RFE): SVM models are powerful classification tools aiming to find a hyperplane that maximizes the distance between binary labelled observation samples [41]. We applied the standard nonlinear SVM with the radial basis function (RBF) kernel embedded with recursive feature elimination (RFE) on our dataset. We give further details on the embedded RFE algorithm for feature selection in the next subsection. We used the Libsvm package to apply the SVM-RFE models [42].Gaussian process classification model: Gaussian process classification (GPC) models are a class of machine learning models which are based on non-parametric Bayesian formulation. In GPC settings, a latent variable f∈R that represents the classification logit is defined and a prior distribution is placed over the latent space in the form of a Gaussian process (GP) [43,44]. We used the Gaussian Processes for Machine Learning (GPML) toolbox to implement the GPC model training in this study [45]. We chose a linear mean function as the prior function of GP model and used the square exponential function for the covariance function.Gradient Boosting Decision Tree: Gradient boosting decision tree (GBDT) is an ensemble model of decision trees in which each decision tree is sequentially built on the gradient descent direction of a loss function. In each iteration, GBDT learns the decision trees by fitting the negative gradients known as residual errors [46].In this paper, we used the software library, eXtreme Gradient Boosting (XGBoost), which is an implementation of GBDT in Python designed for speed and performance [47]. Tuning the XGBoost can be a very daunting task because of the number of hyperparameters it has. We applied grid search with reasonable ranges on only two of the parameters, the number of trees and the maximum tree depth. All the possible combinations of these two parameter values are run for the model tuning and the one with best performance is retained as the optimal values. The rest of the parameters were kept as default in XGBoost library. The final values for the number of trees and the maximum tree depth were set to 4 and 3, respectively.

$D_{sep,t}$ was input to the recurrent neural network (RNN) models since they are able to capture the time-varying nature of the data [48]. These networks have been demonstrated to be useful for learning sequences containing long-term patterns, due to their ability to maintain long-term memory. Long short-term memory models (LSTMs) are a particular kind of RNNs that were introduced to overcome the problem of vanishing/exploding gradients in RNNs by employing multiplicative gates that enforce constant error flow through the internal states of special units called the memory cells [48].

We used LSTMs for binary classification using the Keras package in Python [49]. Our deep learning model architecture comprises an LSTM layer with 24 units with a “sigmoid” activation function. All models are trained on batches of 10 samples with the binary cross-entropy criterion, using “adam” optimization with the default learning rate of 0.01.

#### 3.4.3. Machine Learning Interpretation

For each trained ML model, we applied interpretable analysis within the context of the model. The aim was to quantify the ranking and contribution of each single feature in the final prediction performance. To achieve this, we applied the following methods for each classifier.

RFE for SVM classifier: RFE is an embedded feature selection method based on a backward sequential selection that eliminates a feature in a feature set of size *m* that has the least effect on the SVM weight-vector norm at each iteration [50]. This way, the features are ranked and the SVM classification is repeated *m* times while the last ranked features are removed. Finally, a subset of features with size *r* that optimises the performance of the SVM classifier are selected.GP interpretability framework: We applied a recently developed interpretability analysis of GPC models, based on an explicit form of the GP inference equations to quantify the importance of each feature contributing to the GPC model prediction [51]. Within this framework, small perturbations are propagated to each data input in succession through the prior model and then the GP posterior, in order to quantify the contribution of each feature input to the overall model prediction of a data sample. In particular, given a GP model trained on a given dataset, a test input point and a neighbourhood around the latter, we compute the probability that there exists a point in the neighbourhood such that the prediction of the GP on the latter differs from the initial test input point by at least a given threshold. The outcome is an interpretability metric, denoted Φ, that corresponds to the importance of each data point in the model training.Let us define $x\in R^{n}$ as a generic input data sample in our dataset where $x_{l}$ is the sub-vector of *x* that includes only the indices of l⊆{1,2,…L} and *L* is the total number of features. For any test sample data $x^{*}$ with a subset of indices *l*, a norm |·|, and a radius γ>0, we perform a set of perturbations of magnitude up to γ around $x^{*}$ according to Equation (Equation 2).
(2)$$T_{\gamma ,x^{*}}^{l}=\{x\in R^{n}\;s.t.\;|x_{I}-x_{l}^{*}\left|\leq \gamma \right\}$$Then, we define the interpretability metric $\phi (T_{\gamma ,x^{*}}^{l})$ according to Equation (Equation 3) which reflects how much local perturbations of the indices *l* of $x^{*}$ can change the prediction probabilities.
(3)$$\phi \left(T_{\gamma ,x^{*}}^{l}\right)=\max_{x\in T_{\gamma ,x^{*}}^{l}}\pi \left(y=1|D,x\right)\;-\min_{x\in T_{\gamma ,x^{*}}^{l}}\pi \left(y=1|D,x\right)$$
where π(y=1|D,x) encodes the probability that *x* belongs to class 1. Detailed mathematical formulation of this framework can be found in [51,52].Feature importance in GBDT: Decision trees bring the benefit of interpretability by means of decision analysis on the structure of the trees. One of the main features of GBDT algorithms is that they identify attributes that contribute the most towards the performance. We quantified the importance of each feature based on the number of times a feature is used to split the data across all trees.Within XGBoost library, a feature importance score can be obtained based on the relative contribution of the corresponding feature to the model calculated by taking each feature’s contribution for each tree in the model [46]. A higher value of this metric when compared to another feature implies it is more important for generating a prediction. We selected the “gain” value to report the feature importance results.Interpretable model-agnostic explanation for LSTM: In recent years, among the deep-learning methods, local interpretable model-agnostic explanations (LIME) [53] has emerged as a new evaluation method that can explain the predictions of any classifier by approximating it locally with an interpretable model. It builds a local linear approximation of a complex model’s behaviour in the neighbourhood of a data sample by treating the model as a black box and classifying near permutations of the data sample being explained. Therefore, the output of LIME is a list of explanations, reflecting the contribution of each feature to the prediction of a data sample. This provides local interpretability and allows the determination of the features with the most important impact on the prediction of the data sample.For tabular data, variations of the data are produced by perturbing each feature individually. In particular, we applied the TabularLIME algorithm from the LIME package in Python to quantify the importance of each feature at each timestamp for the trained LSTM model.

## 4. Results

### 4.1. Results with Static Features

We report the results of ML models trained on $D_{sep}$ within two cross-validation schemes suitable for imbalanced datasets. The first one is the leave-one-subject-out (LOSO) cross-validation scheme, in which at each training iteration one patient is left out as a test set while the rest of the patients form the training set. The second method is the stratified K-fold (SKfold). Within this scheme, the dataset is split into k consecutive folds, where each fold is construed to preserve the percentage of samples for each class and is used as the test set while the remaining k − 1 folds form the training dataset. In both cases, the final prediction vector is constructed by concatenating all of the predictions at each iteration.

We report the performance of each classifier within the two selected cross-validation schemes by calculating the performance metrics which are known to be suitable for imbalanced prediction problems.

Precision (PPV): The percentage of truly positive predictions out of the positive predicted.F1-score: Harmonic mean of precision and recall where recall is the percentage of predicted positive out of the total positive. This metric takes both false positive and false negatives into account.AUCROC: The area under the curve of receiver operating characteristic curve.AUCPRC: The area under the curve of precision recall curve.

These results are reported in Table 3 for each feature set for the SVM, GP, and XGBoost classifiers with LOSO and K-fold cross-validation schemes. All the performance metrics are consistent (with marginal differences) between LOSO and K-fold cross-validation schemes, which adds to the generalizability power of the trained models. Moreover, using all the models, the feature sets $Sep_{HRV}$ or $Sep_{HRVital}$ resulted in higher performances compared to the $Sep_{Vital}$ feature set. The SVM-RFE model achieved the highest PPV (96.15%), F-score (86.11%), AUCROC (0.89), and AUCPRC (0.74) within LOSO cross-validation using both HRV and vital signs. Although these metrics are marginally lower when considering only the HRV features, they drop significantly when only the vital signs are considered.

With GP models, although the results are lower compared to the SVM models, the highest performances were obtained using only the HRV features, 88.46% of PPV, 80.50% of F-score, 0.78 of AUCROC, and 0.67 of AUCPRC. This trend is also true for the XGBoost model except for PPV of 92.3% with $Sep_{Vital}$ within K-fold cross-validation which is higher than the PPV achieved with $Sep_{HRV}$ and $Sep_{HRVital}$. However, with XGBoost models, the AUCROC and AUCPRC metrics in all feature sets are much lower than SVMs and GP models.

To obtain an explanation for the behaviour of each classifier we quantify the contribution of each feature in the final prediction of each test data. We illustrate this for SVM-RFE, XGBoost, and GP classifiers in terms of heatmaps in Figure 4. These heatmaps are obtained within the LOSO cross-validation scheme in order to obtain an explanation of the prediction of each trained model at single-patient level. The *x* axis in these figures represents the feature index in each feature set and the *y* axis represents the patient index. The feature contribution values are normalized between (0, 1) and illustrated in colours according to the colour bar. Interestingly, in the heatmaps obtained from GP and XGBoost models, the contribution values are consistent within different patients. However, the SVM classifier seems to offer a more personalised feature ranking for each patient. The features which contributed most during GP training for the majority of patients are HR (mean) and SD2 (Figure 4d), while for XGBoost, these are HR (mean), RR (mean), and $Peak_{LF}$ (Figure 4g). For the SVM, HR (mean) is also among the first ranking feature for all the patients (Figure 4a).

Among the features in $Sep_{Vital}$, SVM ranks SBP as the highest contributing feature among all vital signs (Figure 4b). However, Pulse and Resp show the highest contribution in the GP model (Figure 4e), while for XGBoost, only Pulse is used for its training (Figure 4h). This observation for XGBoost is the same when the HRV features are combined with vital signs (Figure 4i), while for the GP model, the SBP, SP02 from the vital signs, and some HRV features show high contributions (Figure 4f).

### 4.2. Results with Time-Varying Features

We report the results of the LSTM models trained on $D_{sep,t}$ using the group stratified cross-validation scheme. Through this method, stratified folds are considered with non-overlapping groups while preserving the percentage of samples for each class. We report the performance of LSTM models with AUCPRC and AUCROC metrics in Table 4. Both performance metrics are the highest (AUROC of 70% and AUCPRC of 83%) considering the $Sep_{HRV}$ feature set. Due to the time-varying nature of the dataset, we also considered a new univariate feature set comprising only the mean of heart rate, which resulted in relatively high performance (AUROC of 68% and AUCPRC of 82%).

We applied the LIME algorithm to the trained LSTM models to obtain a single patient level explanation. We considered the three feature sets with the highest performances ($Sep_{HRV}$, $Sep_{HRVital}$, HR(mean)) in Table 4 and applied LIME for exemplary test data that were predicted to belong to the non-survival group using all the feature sets (Figure 5). In these figures, we illustrated the first 10 highest contributing features to the prediction at the specific timestamp. Therefore $f_{t-j}$ represents the feature value at the timestamp t−j where j⊆{1,2,…T−1} and *T* is the length of observation time in hours for each feature, which is 24 h in this study. The x-axis shows the relative contribution of a feature value at a specific time with the highest contributing feature at the top. The y-axis shows the value of each feature at a specific time stamp. The bar lines on the positive side of the axis reflect the positive effect on the prediction whereas the bar lines on the negative side of the axis reflect the negative effect of that feature on the prediction.

## 5. Discussion

The experimental results using both time-varying dynamics of features and their averaged value show the higher performance of the $Sep_{HRV}$ and $Sep_{HRVital}$ feature sets which include the data from the wearable sensor compared to the $Sep_{Vital}$ feature set which includes data collected only from bedside monitors. Since the dataset is imbalanced, the AUCPRC is a more suitable performance metric to compare the prediction performances, which is the highest (0.83%) considering the time varying features from $Sep_{HRV}$ feature set. However, with a marginal difference in AUCPRC (only 1%), the time varying dynamic of the HR offers informative information in predicting the mortality of sepsis patients. This is reflected also in the heatmaps obtained from static features. HR is the first ranking feature selected by SVM for the majority of patients in $Sep_{HRV}$. It is the feature with the highest contribution among the features in $Sep_{HRV}$ for prediction considering the GP models. However, the best performances are achieved by considering a combination of all HRV features. Considering the Xgboost performance in Table 3 and the relevant heatmaps in Figure 4, it is evident that Xgboost is using only the *Pulse* feature in $Sep_{vital}$ and $Sep_{HRVvital}$ feature sets for prediction (Figure 4h,i), which could be the main reason why it is leading to a worse performance compared to the combination of HRV features ($Peak_{LF}$, HR (mean), and RR (mean) used from $Sep_{HRV}$ (Figure 4g).

Considering the time varying features, the local explanation of the LSTM models through the LIME algorithm leads to a better understanding of the behaviour of these models by knowing which features at which timestamps make the most contributions to the prediction of the patient mortality. It is particularly important for clinicians to know not only which physiological factors are the most informative but also to identify which timestamps are the most important for a individual patient’s monitoring. Using wearable sensors, we were able to monitor patients over a 24 h period, shortly after admission, and these features would have been missed if we used only short 10 or 30 min recordings from very early on. Within this time window, it is possible that our models are detecting patients’ treatment responses, and therefore could be particularly valuable in resource limited settings with few staff available.

As we can see from the exemplary LIME analysis of a non-survived sepsis patient in Figure 5, the time stamp 0—which is the last hour of the monitoring—is among the highest contributing factors using $Sep_{HRV}$ and $Sep_{HRVital}$ features. However, if we consider only the HR as a mortality predictor, the 12th h after the admission is the most important moment for this exemplar patient. Interestingly, the features *S* (area of the fitted ellipse in Poincare plot) and $P_{HF}$ (absolute power in HF band) are among the first two highest contributing features to the model prediction in $Sep_{HRV}$ and $Sep_{HRVital}$ feature sets.

The presence of HRV features in the first rankings, considering the $Sep_{HRVital}$ feature set in SVM models, the outperformance of $Sep_{HRV}$ feature set from GP and XGboost models with respect to the other two feature sets, and the highest AUCPRC achieved from the time varying $Sep_{HRV}$ feature set, are all strong reasons to consider the data collected from wearable sensors instead of from bedside monitors for the effective monitoring of sepsis patients. This is because the HRV dynamics capture the autonomic nervous system function and therefore reflect the activity of physiological compensatory mechanisms which may be significant even though vital signs remain unchanged. For this reason, HRV monitoring using wearable sensors (as the most practical approach) could give valuable additional insight into the trajectory of patients with sepsis and their response to treatment. Even if the SVM model performed better with the $Sep_{HRVital}$ feature set compared to $Sep_{HRV}$ using static features, the difference is so marginal that relying only on data gathered from wearable monitoring systems is still justified. Apart from the device’s performance, the cost implications and the shortage of expert healthcare staff resource are particularly paramount in LMIC settings, which further justify the use of wearable devices in these healthcare systems.

We reported our results within different cross-validation schemes suitable for each ML model to avoid the overfitting of the models due to the limitations of having a larger small sample size in real life LMIC settings. Although the generalizability of the results can definitely be improved by increasing the sample size of the patient cohort in our study, to the best of our knowledge, this is the first prospective study in an LMIC setting to collect a relatively large dataset of high-resolution ECG signals for long time duration (24 h) from sepsis patients, and from this data propose ML-based prediction algorithms for a better care management of these patients. Another study [19] with a similar data collection protocol of sepsis patients in Rwanda performed statistical analysis with data from 43 patients. This attests to the fact that data collection for research purposes from critically ill patients from a critical unit for research purposes is a very cumbersome task, particularly in low-resource settings.

Another limitation of our study is the imbalanced nature of our dataset (35% for the positive class), which limits the reliance on many performance metrics in supervised learning. However, the task of mortality prediction in clinical practice is often performed with imbalanced datasets and usually with a 5–10% mortality rate. We tried to overcome this limitation by relying on those performance metrics (e.g., AUCPRC) that are less affected by the low proportion of positive cases in the dataset.

Unlike many other studies, we have not applied extensive preprocessing methods on the ECG signals collected from the ePatch. This choice was firstly to reduce the processing computational costs as much as possible, and secondly to make our prediction pipeline suitable for real-life scenarios, rather than controlled lab conditions. Due to the very low computational cost of the training models used in this study, we believe there is a great potential to directly deploy these models in clinical practice. In fact, in our future studies we aim to implement the proposed decision systems in clinical practice for a more efficient method of monitoring critically ill sepsis patients in the HTD hospital. Moreover, to overcome the limitations caused by sample size, we aim to take advantage of similar datasets with a much larger sample size collected in high-income settings and use transfer learning techniques to generalise the results to our smaller dataset.

## 6. Conclusions

In this study, we presented an interpretable automatic ML-based pipeline for the long-term prediction of mortality in sepsis patients in the Vietnamese hospital. We demonstrated the practicality of wearable sensor monitors in monitoring patients with sepsis in LMIC settings. Our results show that the information extracted from the ECG signal acquired from low-cost wearable sensors results in higher performances compared to the information collected from expensive bedside monitors, for mortality prediction of sepsis patients. The highest performed prediction model was the LSTM model, using the time varying dynamics of HRV indices. With LSTM models, an AUCPRC of 0.83 was achieved using the HRV features extracted from the 24 h ECG signals of sepsis patients after hospital admission.

We provided interpretability analysis for all of the ML prediction models at a single patient level, showing the high contribution of HRV-based features leading to mortality prediction. The interpretability analysis of the SVM and Gaussian process models show that the feature HR(mean) is the most informative feature in the $Sep_{HRV}$ feature set for mortality prediction of critically ill patients (Table 5). Moreover, through the LIME analysis of LSTM models, the features derived by nonlinear analysis of HRV (e.g., area of the flitted ellipse in Poincare plot) contributed the most to the prediction task.

Our study is among the few studies that have conducted wearable monitoring and proposed an automatic mortality prediction pipeline within an LMIC setting in critically ill patients. The main challenges of our study are the relatively small sample size of the patients and the lack of external validation in the clinical setting. However, the findings of this study are helpful to expand research in utilizing wearable technology integrated with ML-based prediction models in managing infectious diseases in hospitals in resource limited settings.

## Figures and Tables

**Figure 1 sensors-22-03866-f001:**
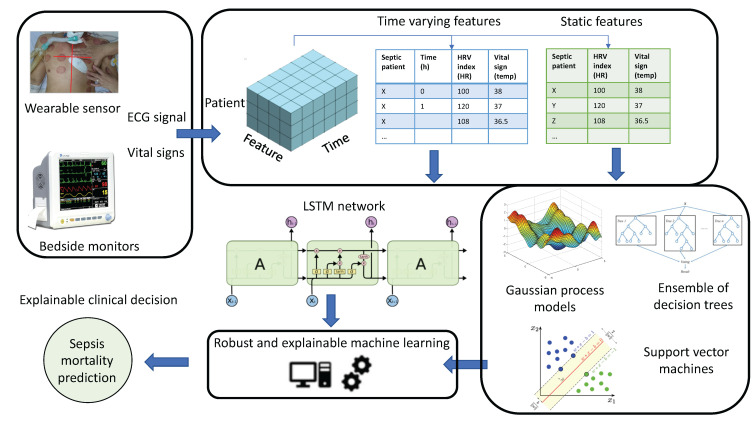
Processing pipeline in this study.

**Figure 2 sensors-22-03866-f002:**
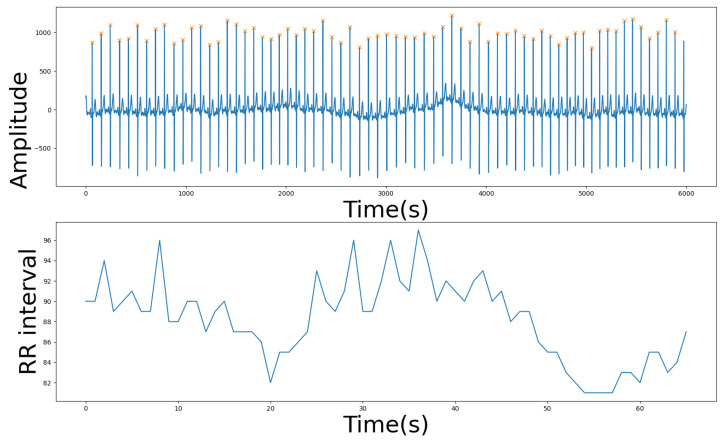
Sample 1 min ECG recording from ePatch^®^ (**top** figure) and the corresponding RR interval (**bottom** figure).

**Figure 3 sensors-22-03866-f003:**
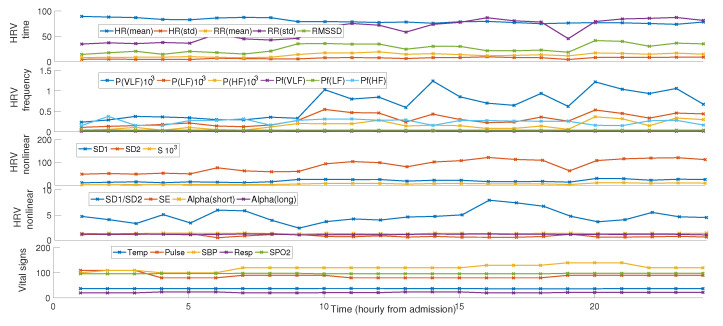
Dynamic trend of the features used in this study for an exemplar patient.

**Figure 4 sensors-22-03866-f004:**
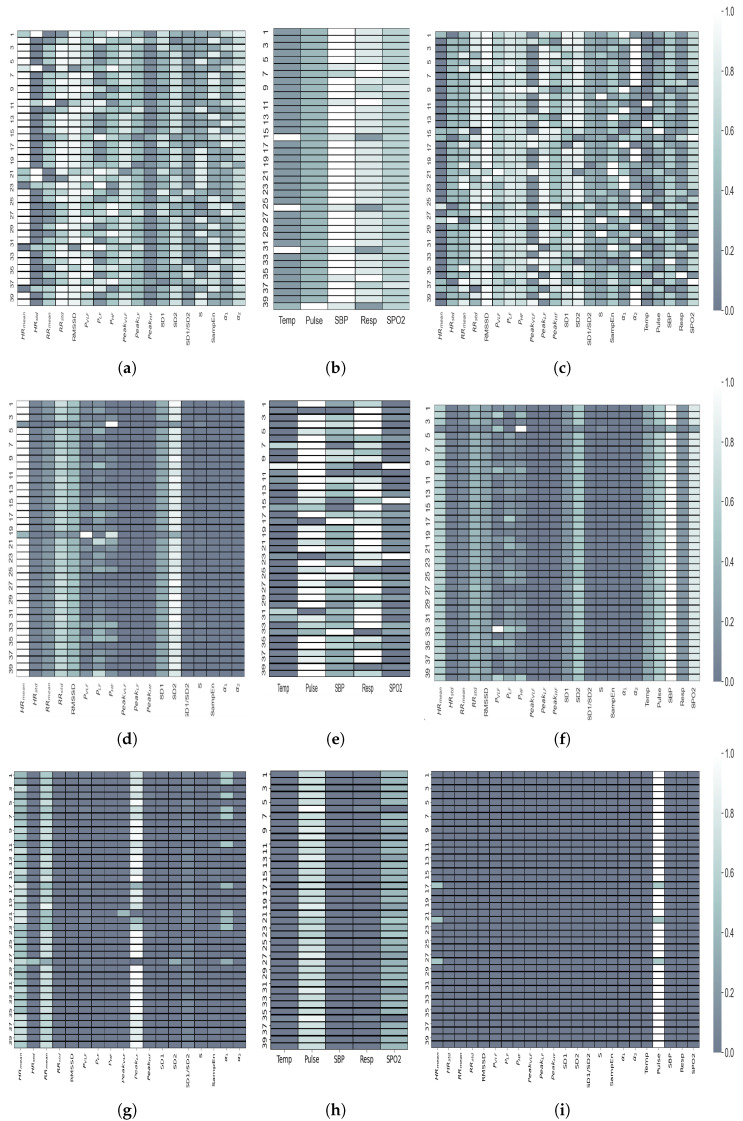
Heatmaps representing the contribution of each feature for the SVM (top), Gaussian process (middle), and XGBoost (bottom) models for each input feature set. The y-axis shows the patient index and the x-axis is the feature names in each feature set. (**a**) HRV features, (**b**) vital signs, (**c**) HRV and vital signs, (**d**) HRV features, (**e**) vital signs, (**f**) HRV and vital signs, (**g**) HRV features, (**h**) vital signs, (**i**) HRV and vital signs.

**Figure 5 sensors-22-03866-f005:**
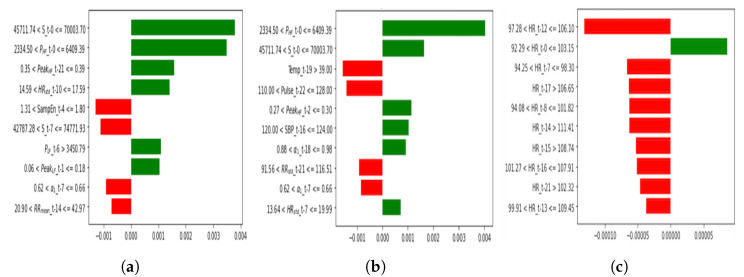
Local explanation of the LSTM models for a non-survival test sample based on LIME analysis. The green bar lines to the right for a feature at time *t* reflect the positive effect of that feature for the test sample to be assigned to the non-survival class while the red bar lines to the left show the opposite. (**a**) HRV features, (**b**) HRV features and vital signs, (**c**) heart rate.

**Table 1 sensors-22-03866-t001:** Demographic information of the sepsis patient population included in this study.

Variable	All (n = 40)	Death (n = 14)	Survival (n = 26)
gender (M)	67.5 %	64.3 %	72 %
Age (>=64)	n = 12	n = 2	n = 10
Age (50–64)	n = 7	n = 1	n = 6
Age (<50)	n = 21	n = 11	n = 10
Hospital length of stay	12.08 ± 12.25	10.14 ± 15.97	13.04 ± 9.9
SOFA at admission	2.13 ± 1.74	1.77 ± 1.58	2.78 ± 1.89

**Table 2 sensors-22-03866-t002:** List of extracted HRV features in this study.

Parameter	Unit	Description
**HRV time parameters**		
HR(mean)	BPM	Mean of heart rate
HR(std)	BPM	Standard deviation of heart rate
RR(mean)	ms	Mean of RR intervals
RR(std)	ms	Standard deviation of RR intervals
RMSSD	ms	Root mean square of successive
		RR interval differences
**HRV frequency parameters**		
$P_{VLF}$	ms^2^	Absolute power in VLF band
$P_{LF}$	ms^2^	Absolute power in LF band
$P_{HF}$	ms^2^	Absolute power in HF band
$Peak_{VLF}$	Hz	Frequency where maximum power
		occurs in VLF band
$Peak_{LF}$	Hz	Frequency where maximum power
		occurs in LF band
$Peak_{HF}$	Hz	Frequency where maximum power
		occurs in HF band
**HRV nonlinear parameters**		
SD1	ms	Standard deviation along the minor axis
		in Poincare plot
SD2	ms	Standard deviation along the major axis
		in Poincare plot
SD1/SD2	-	Ratio between SD1 & SD2
S	-	Area of the fitted ellipse (Poincare plot)
SampEn	-	Sample entropy of RR series
$\alpha_{1}$	-	Alpha value of the short term fluctuations
		in detrended fluctuation analysis
$\alpha_{2}$	-	Alpha value of the long term fluctuations
		in detrended fluctuation analysis
**Vital signs**		
Temp	°C	Temperature
Pulse	BPM	Hear rate
SBP	mmHG	Systolic blood pressure
Resp	BPM	Respiratory rate
SP02	%	Peripheral capillary oxygen saturation

**Table 3 sensors-22-03866-t003:** In-hospital mortality prediction results in sepsis patients using static features.

	*SVM-RFE*			*Gaussian Process*			*XGBoost*		
	${\mathrm{Sep}}_{\mathrm{HRV}}$	${\mathrm{Sep}}_{\mathrm{Vital}}$	${\mathrm{Sep}}_{\mathrm{HRVital}}$	${\mathrm{Sep}}_{\mathrm{HRV}}$	${\mathrm{Sep}}_{\mathrm{Vital}}$	${\mathrm{Sep}}_{\mathrm{HRVital}}$	${\mathrm{Sep}}_{\mathrm{HRV}}$	${\mathrm{Sep}}_{\mathrm{Vital}}$	${\mathrm{Sep}}_{\mathrm{HRVital}}$
PPV (LOSO)	92.31	84.62	96.15	88.46	80.77	80.77	84.62	76.92	76.92
PPV (SKfold)	92.31	84.62	92.31	84.62	76.92	80.77	84.62	92.31	76.92
F1-score (LOSO)	86.09	74.90	86.11	80.50	69.31	72.53	77.31	72.70	72.70
F1-score (SKfold)	86.09	78.02	86.09	78.02	60.30	69.31	79.00	79.00	75.33
AUCROC (LOSO)	0.80	0.76	0.89	0.78	0.69	0.73	0.67	0.60	0.60
AUCROC (SKfold)	0.83	0.78	0.80	0.77	0.68	0.72	0.70	0.68	0.67
AUCPRC (LOSO)	0.71	0.64	0.74	0.67	0.48	0.58	0.49	0.35	0.38
AUCPRC (SKfold)	0.72	0.65	0.74	0.67	0.46	0.59	0.55	0.55	0.43

**Table 4 sensors-22-03866-t004:** In-hospital mortality prediction in sepsis patients using time varying features and LSTM.

	${\mathrm{Sep}}_{\mathrm{HRV}}$	${\mathrm{Sep}}_{\mathrm{vital}}$	${\mathrm{Sep}}_{\mathrm{HRVital}}$	HR(mean)
AUCROC	**0.70**	0.62	0.67	0.68
AUCPRC	**0.83**	0.72	0.81	0.82

**Table 5 sensors-22-03866-t005:** Highest contributing features in the final outcome prediction of each ML model for each feature set.

	${\mathrm{Sep}}_{\mathrm{HRV}}$	${\mathrm{Sep}}_{\mathrm{vital}}$	${\mathrm{Sep}}_{\mathrm{HRVital}}$
SVM	$HR_{mean}$	SBP	RMSSD
Gaussian Process	$HR_{mean}$	Pulse	SBP
XGBoost	$Peak_{LF}$	Pulse	Pulse

## Data Availability

Data not publicly available due to ethical restrictions.

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
