# Peer review of "Sepsis Mortality Prediction Using Wearable Monitoring in Low–Middle Income Countries"

_sensors, 2022, doi:10.3390/s22103866_

Round 1
Reviewer 1 Report
The paper assesses the feasibility of using wearable sensors instead of traditional bedside monitors in the sepsis care management of hospital admitted patients and introduces automated prediction models for the mortality prediction of severe sepsis patients. The paper has several technical and methodological issues, which must be resolved before the paper could be considered for publication.
Comments:
- Avoid mass citing such as [2-6], but use each reference separately to support a particular item or claim.
- Describe your contribution in a more specific way. What are your innovations? How is your work different from other similar studies?
- The analysis of related works is mediocre. There is no introductory paragraph. Improve the discussion on the methods for extraction of physiological signals. For example, you can discuss doi:10.1016/j.bspc.2020.101873, doi:10.1007/978-3-030-81473-1_3.
- Present a motivation for the specific design decisions you have taken: “an LSTM layer with 24 units with a "sigmoid" activation function”. Why 24 layers? Why sigmoid activation function was used? Can you present the results of the ablation study? Present the settings of the hyperparameters such as learning rate and batch size.
- Present the architectural diagram of the LSTM model.
- The details in Figure 4 are too small to be readable. How the results are explained?
- Present the plots for the ROC curve and the PR curve, and discuss.
- What is the statistical significance (p-value) of the effects shown in Figure 5?
- I do not find the contribution towards the explainability of the model results. You should be able to demonstrate which features and which values of the feature motivated the clinical decision.
- Improve the conclusions, and use the main findings from the experiments and statistical analysis to support your claims.
Reviewer 2 Report
It is a well-motivated study for improving the healthcare monitoring in low-middle income countries. The wearable sensors were used to detect the heart-rate-variability (HRV) signals for sepsis mortality prediction. The following comments could be considered for manuscript improvement:
1) In Figure 2, it can be observed that the motion artifacts are present in the ECG recording. The signal preprocessing procedure seems necessary before the HRV analysis.
2) Is the training implemented real-time or offline? Please describe the training data source and strategy.
3) Some parameters of the machine learning models should be provided, such as the kernels of the LibSVM, the parameters of Gaussian process. The settings of XGBoost and the architecture details of the LSTM should also be described.
Reviewer 3 Report
The article reports a Machine learning (ML) technique for wearable mechatronics systems to predicate sepsis morality. The manuscript is well written and organized; however, some modifications are required.
- why authors have used ML methods ? an explicit explanation is required.
- challenges of this work should be reported in the conclusion section.
Round 2
Reviewer 1 Report
The authors have revised well. All my comments were addressed. I have no further comments and recommend to accept.